# Pregnane X receptor activation constrains mucosal NF-κB activity in active inflammatory bowel disease

**J. Jasper Deuring, Meng Li, Wanlu Cao, Sunrui Chen, Wenshi Wang, Colin de Haar, C. Janneke van der Woude, Maikel Peppelenbosch** *

Department of Gastroenterology and Hepatology, Erasmus MC-University Medical Center, Rotterdam, The Netherlands

* m.peppelenbosch@erasmusmc.nl

**Data Availability Statement:** All relevant data are within the manuscript and its Supporting Information files.

## Abstract

### Background

The Pregnane X Receptor (PXR) is a principal signal transducer in mucosal responses to xenobiotic stress. It is well-recognized that inflammatory bowel disease is accompanied by xenobiotic stress, but the importance of the PXR in limiting inflammatory responses in inflammatory bowel disease remains obscure at best.

### Methods

We stimulate a total of 106 colonic biopsies from 19 Crohn's disease patients with active disease, 36 colonic biopsies from 8 control patients, colonic organoids and various cell culture models (either proficient or genetically deficient with respect to PXR) *in vitro* with the PXR ligand rifampicin or vehicle. Effects on NF-κB activity are assessed by measuring interleukin-8 (IL-8) and interleukin-1ß (IL-1ß) mRNA levels by qPCR and in cell culture models by NF-κB reporter-driven luciferase activity and Western blot for signal transduction elements.

### Results

We observe a strict inverse correlation between colonic epithelial PXR levels and NF-κB target gene expression in colonic biopsies from Crohn's disease patients. PXR, activated by rifampicin, is rate-limiting for mucosal NF-κB activation in IBD. The correlation between colonic epithelial PXR levels and NF-κB target gene expression was also observed in intestinal organoids system. Furthermore, in preclinical *in vitro* models of intestinal inflammation, including intestinal organoids, genetic inactivation of PXR unleashes NF-κB-dependent signal transduction whereas conversely NF-κB signaling reduces levels of PXR expression.

### Conclusions

Our data indicate that the PXR is a major and clinically relevant antagonist of NF-κB activity in the intestinal epithelial compartment during inflammatory bowel disease.

**Funding:** M.L. [201506100033] and S.C. [201606760056] are supported by a China Scholarship Council stipend (https://www.chinesescholarshipcouncil.com/), M.P. receives funding from the Dutch Society for the Replacement of Animal Testing and ZONMW (2016/22827/ZONMW) for the financial support of this work (www.zonmw.nl). The sponsors were not involved study design, data collection and analysis, decision to publish, or preparation of the manuscript.

**Competing interests:** The authors have declared that no competing interests exist.

## Introduction

Intestinal epithelial cells (IEC) form the physical barrier between the gut content and the *milieu interieur* and perform a multitude of functions in cellular physiology including absorption of nutrients and water but also constitute a first line of defense against pathogenic and xenobiotic challenge to the body [1, 2]. The interaction between IEC functionality in innate immunity and xenobiotic detoxification remains largely obscure but is likely relevant in pathophysiology as pathogenic and xenobiotic stress often occurs concomitantly in the intestine [3], and breakdown of barrier function by specific epithelial subtypes underpins inflammatory bowel disease (IBD) [4]. Xenobiotics are often the result of metabolism by specific bacteria, which fits well with the insight that altered microbial composition is linked to the clinical course of IBD [5] as well as reaction to therapy [6]. Various receptor systems are involved in the detection by IEC of xenobiotic components present in the in the intestinal lumen, in particular the plasma membrane-localized G-protein-coupled receptors GPR41, GPR43, and GPR109A and the nucleus-localized receptors aryl hydrocarbon receptor, farnesoid X receptor and pregnane X receptor (PXR) [7]. With respect to the nuclear receptors, the aryl hydrocarbon receptors protects stem cells against challenge to their genome by genotoxic compounds through stimulating the production of interleukin 22 by lymphocytes [8] from the diet, whereas generally speaking this receptor has a regulatory influence on immunity through Src-mediated stimulation of indoleamine 2,3-dioxygenase 1 [9], an enzyme that is a key element in relay pathway between arginine and tryptophan metabolism that mediates immunosuppression [10]. For other xenobiotic-sensing nuclear receptors in general and PXR in particular, their potential functionality in limiting intestinal inflammation is less clear-cut.

Intriguingly, however, the PXR locus is associated with susceptibility to IBD, suggesting that this receptor is clinically relevant in constraining in intestinal inflammation [11, 12]. In apparent agreement, stimulating PXR in rodents during experimental colitis ameliorates inflammation and reduces disease [13–17]. Mechanistically these effects may relate to intestinal NF-κB activation. NF-κB is a master regulator of inflammatory responses of the genome [18] and its importance for the pathogenesis for inflammatory bowel disease is undisputed [19]. Importantly PXR deficient mice display more severe NF-κB-driven small intestinal inflammation than their non-mutant littermates [20] whereas effects of PXR activation in experimental colitis also correlate to NF-κB activation [21]. It thus rational to propose that also in human disease, PXR activation constrains NF-κB activation and IBD. The functionality, however of PXR activation in clinical inflammatory bowel disease and its relation to NF-κB activation remains, however, largely unexplored.

Prompted by the above-mentioned considerations we decided to explore the role of PXR activation in human IECs and clinical IBD. Our study shows that PXR activity is the major rate-limiting pathway constraining mucosal NF-κB activity in active IBD and provides insight into PXR signals, which are much more important in pathology than previously thought. Furthermore, our results imply that modulation of PXR activity holds significant clinical promise in the management of IBD.

## Materials and methods

### Cell lines

All cell lines were originally obtained from the ATTC. Human colorectal adenocarcinoma cell lines CACO2 and LS174t and hepatocellular carcinoma cell lines Huh7 were cultured in Dulbecco's modified Eagle's medium from Invitrogen-Gibco, complemented with 10% fetal calf serum, 100 IU/ml penicillin and 100 ug/ml streptomycin according to routine procedures

[22]. Cell lines were used to investigate the function of PXR expression. All cell lines needed to passage twice a week and were cultured according standard culture conditions.

## Gene knockdown

To study the NF-κB inhibiting potential of PXR activation, a PXR knockdown LS174t cell was created to test the specificity of PXR. The LS174t cell line was transduced with the lenti-virus, containing small interference RNA for PXR (siPXR), similarly as described before [23]. The transduced cells were cultured with 100 uM puromycin (Sigma-Aldrich) for three weeks to select for cells that harbor the siPXR.

## Reagents

NF-κB was in-vitro activated by 2 μl *E. coli* lysate (ELI), a centrifuged 50 ml o/n *E. coli* (DH5alfa, Invitrogen) culture taken up in 500 ul dH$_2$O. PXR was activated by 100 μM Rifampicin (Sigma-Aldrich). Recombinant human TNFα (Perotech, USA) was dissolved in phosphate-buffered saline in stock solution of 100 μg/ml.

## Biopsies

This study was conducted with approval of the Ethic committee of the Erasmus MC University Medical Center in Rotterdam. All patients gave written informed consent. During endoscopy biopsies were taken from patients with a known history at least 6 months of CD and from patients referred for colonoscopy but without intestinal abnormalities, further described as control patients. Also patients were asked for additional blood samples. CD was diagnosed according to international guidelines and only the results of the control biopsies were used if there were no abnormalities on pathology, when there was no history of IBD, and no familiar history of IBD. For each patient the biopsies were taken from the ascending colon ($n = 3$), the transversum ($n = 3$) and the descending colon ($n = 3$). All information of biopsies was shown in S4 Table.

## Culture of human intestinal organoids

Human intestinal organoids were cultured as described previously [24]. Briefly, intestinal tissues were re-suspended in advanced DMEM/F12 (supplemented with 1% GlutaMAX™ Supplement, 10 mM HEPES) with growth factors, and collected by centrifugation. Crypts were finally suspended in Matrigel (Corning, Bedford, USA), and placed 40 μL/well in a 24-well plate. Organoids were cultured in expansion medium after the Matrigel had solidified. Organoid expansion medium was refreshed every 2–3 days, and organoids were passaged every week.

## Histology

One biopsy from each location was fixed in 4% formaldehyde solution, dehydrated and embedded in paraffin for histological scoring. Four microM slices from the formalin fixed paraffin embedded (FFPE) tissue specimens were stained with hematoxylin and eosin (Sigma-Aldrich) according to previously described procedures [25]. Three observers have independently examined each biopsy, in a blinded fashion. Discrepancies were reassessed to reach agreement.

## Stimulation of the biopsies

The freshly taken biopsies were immediately placed in ice-cold regular culture medium (DMEM) for transport. Before stimulating the biopsies, they were washed three times with ice-cold PBS containing antibiotics to prevent infection. The biopsies were then stimulated for 18 h at 37 °C with 100 μM Rifampicin (Sigma-Aldrich) or solvent. Stimulated biopsies were directly lysed in Tripure (Roche, Switzerland) for RNA and protein extraction, according to the manufacturer's protocol. After the Tripure extraction, the RNA samples were purified using the RNA II extract kit from Macherey Nagel (Bioke) according to the manufacturer's protocol.

## Peripheral blood mononuclear cells

Peripheral blood mononuclear cells (PBMC) were isolated from fresh blood using Ficoll (Gibco) according standard procedures. The isolated PBMC were stimulated with 100 μM Rifampicin for 18 h at 37 °C followed by lysing of the PBMC in Tripure (Roche) for RNA isolation.

## Quantitative real-time polymerase chain reaction (PCR)

Gene expression of GapdH, Ywaz, IL-8, IL-1β, CYP3A4, Sult1a, and PXR were measured via quantitative real-time PCR with the StepOne Real-Time PCR system and the StepOne v2.0 software (Applied biosystem, Darmstadt, Germany). The primer sequences are shown in **S1 Table**. All genes were analysed using the same qPCR program as described before [26]. Gene expression is plotted as fold change using the deltaCt method [27]. The data from patients with multiple colonic biopsies were averaged.

## Luciferase activity measurement of CACO2-based NF-κB luciferase reporter cell lines

Luciferase reporter cells were created by transducing cells with lentiviral vectors expressing the firefly luciferase gene under the control of the NF-κB promoters. The luciferase activity was measured with a LumiStar Optima luminescence counter (BMG Lab Tech, Offenburg, Germany). The cells were cultured and measured as described previously [28, 29].

## Protein analysis

The p-p65 (catalogue no. 3037, Cell Signaling Technology) and p-Akt (catalogue no. 11055–2, Signalway Antibody) protein expression was measured using conventional Western blot as described before [30]. The IL-8 protein expression in the protein solution isolated from the TriPure fraction was measured using ELISA [31], Human IL-8 ELISA Ready-SET-Go! (eBioscience). Immunohistochemistry for NF-κB target genes was performed as described earlier [32].

## Statistics and software

All the graphs and the statistical analyses were performed using the Graphad Prism 5.0 software package for Windows. Data from the paired biopsies were non-parametric statistically analyzed using the Wilcoxon matched pairs test. Correlations were determined using the Spearman's rank correlation coefficient. A two-tailed $P$ value $<0.05$ was accepted as statistically significant. Images were composed using Adobe Photoshop CS5.

### Ethical statement

The work has been approved by the Medical Ethical Committee of the Erasmus Medical Center (Medisch Ethische Toetsings Commissie Erasmus MC), and that subjects gave informed consent to the work.

## Results

### PXR activation is rate-limiting for mucosal NF-κB activation in IBD

To investigate the effects of PXR stimulation on mucosal NF-κB activation, we decided to contrast the effect of the canonical PXR ligand rifampicin [33] to solvent control on NF-κB target gene levels in colonic biopsies. For this experimentation we obtained 106 biopsies from 19 CD patients and 36 biopsies from 8 controls. Demographic patient characteristics are presented in **S2 Table**. Five additional patients with quiescent CD and 4 controls agreed to donate blood samples. We concluded that this set of patient materials should allow us to make meaningful statements on potential effects of PXR stimulation on mucosal NF-κB activation.

The expression of the NF-κB target genes *IL-8* and *IL-1ß* has been shown previously to represent a valid reflection of NF-κB-mediated transcriptional activity [34]. Indeed, when non-stimulated biopsies were investigated for the mRNA levels of these cytokines we observed that expression of either *IL-8* and *IL-1ß* mRNA levels (**Fig 1A and 1B**, respectively) or IL-8 protein levels (**Fig 1C and 1D**, respectively) were markedly higher in biopsies of patients with active inflammation when compared to biopsies from controls or CD patients with quiescent disease. Thus we decided to use expression levels of these two cytokines as a surrogate measure for assessing the effect of PXR stimulation on mucosal inflammation. Importantly, challenge of biopsies with Rifampicin did not significantly reduce the *IL-8* or *IL-1ß* mRNA expression in the biopsies from control and quiescent CD patients (**Fig 1A and 1B**), indicating that outside the context of active IBD, PXR activity is not a rate-limiting factor with respect to NF-κB-directed gene expression. However, Rifampicin stimulation caused a 35 fold reduction of *IL-1ß* mRNA expression in the biopsies with active inflammation ($p<0.01$, **Fig 1B**). Thus PXR stimulation can constrain inflammatory gene expression in active IBD but does not affect constitutive levels of inflammatory cytokines in quiescent IBD or in the colonic mucosa of non-IBD individuals.

### PXR activation limits NF-κB activation in the epithelial compartment

Intestinal PXR expression is especially prominent in the epithelium and is less evident in the stromal and immunological compartment, suggesting that the effects observed following rifampicin stimulation relate to the epithelial compartment [35]. Nevertheless, since the biopsies, especially those of patients with active CD, contain a large number of lymphocytes we wanted to determine if these cells contribute to the observed PXR-mediated reduction of NF-κB signaling. Therefore, we investigated the effect of Rifampicin on PBMC isolated from blood. No *PXR* mRNA were detected in any of the PBMC fractions. Rifampicin stimulation does not influence the *IL-8* mRNA expression in PBMC from controls and CD patients (**S1 Fig**). Furthermore the expression of PXR target gene *Sult1a* is not altered (**S1 Fig**). Hence, mononuclear cells such as stromal cells do not seem to be important in the PXR-mediated inhibition of NF-κB in human intestinal biopsies. This notion was confirmed in experiments in which we tested the effects of rifampicin in TNFα-stimulated human colonic organoids, which are devoid of non-epithelial components. In apparent agreement with the intestinal epithelium being an important mediator of PXR effects, we observed marked reduction of *IL-8* and *IL-1ß* mRNA levels following Rifampicin treatment, whereas such effects were much less

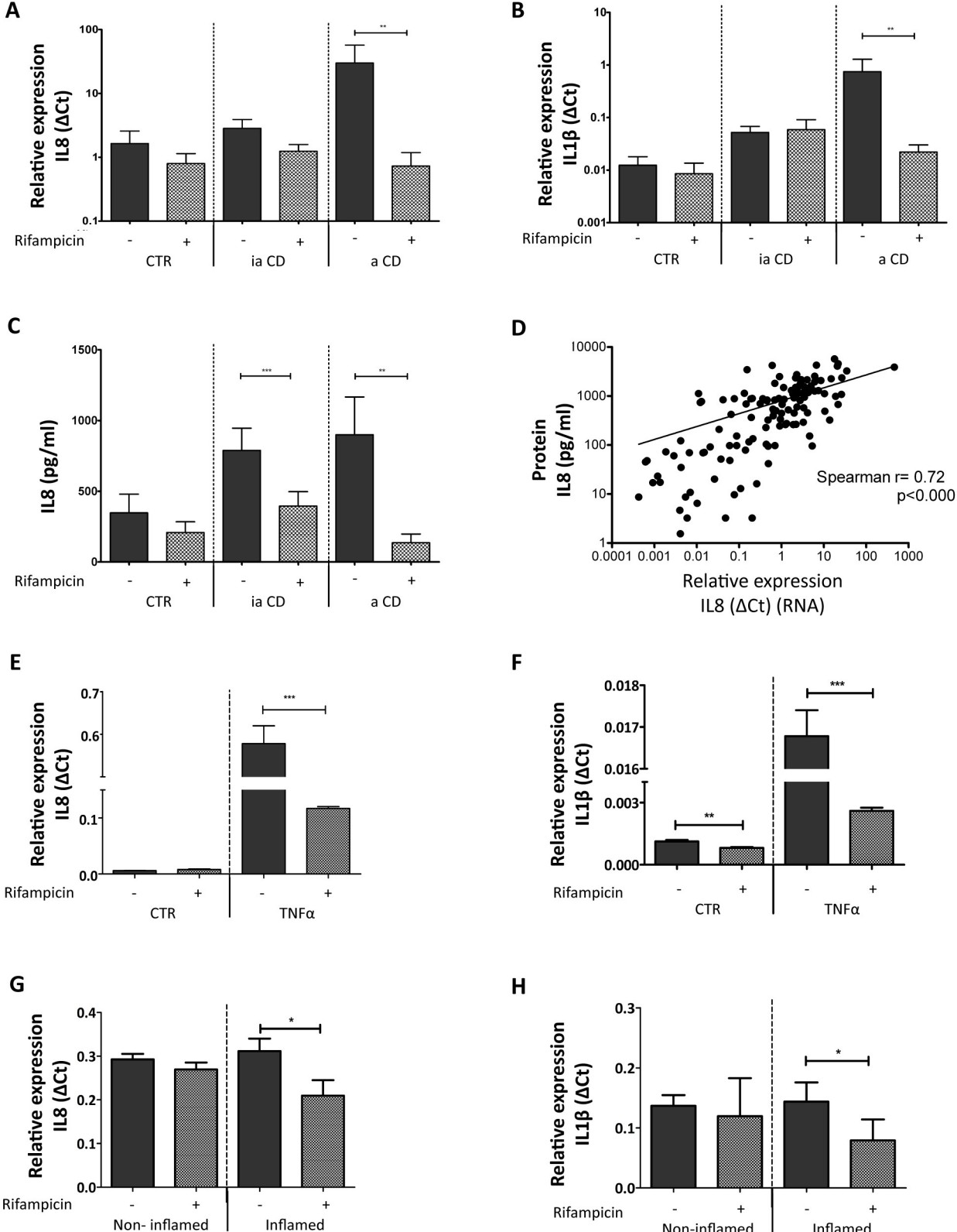

**Fig 1. Effects of with rifampicin treatment on cytokine expression in human intestinal biopsies.** (**A**) IL-8 mRNA expression in human intestinal biopsies. The graph represents the mean IL-8 mRNA expression on a log scale, from biopsies stimulated with solvent only (0.1% (v/v) DMSO) or 100 μM Rifampicin for 18 h at 37 °C. CTR are the biopsies from control patients ($n = 36$), ia CD signifies biopsies from CD patients

without active intestinal inflammation ($n$ = 66), and a CD indicates biopsies from CD patients with active intestinal inflammation ($n$ = 40). (**B**) IL-1βmRNA expression in human intestinal biopsies. For this graph the same labeling applies as in A. The error bar denotes SEM, ** $p$<0.01. (**C**) IL-8 protein expression in human intestinal biopsies. ELISA was used to measure the IL-8 protein concentration from biopsy homogenates. The same biopsies were used as for the mRNA expression analysis in A and B. The error bar is SEM, ** $p$<0.01, *** $p$<0.001. (**D**) Correlation between IL-8 mRNA levels and protein levels. The IL-8 mRNA expression (dCt = δCT) is plotted against the IL-8 protein (pg/mL) measured *per* biopsy. The Spearman correlation is depicted as well, $r$ = 0.72, $p$<0.0001. (**E**) & (**F**) IL-8 mRNA levels and IL-1β levels, respectively, as measured in human intestinal organoids. The graph represents the mean IL-8 mRNA expression stimulated with solvent only (0.1%(v/v) DMSO) or 100 μM Rifampicin for 18 h at 37 °C. The TNFα group was treated with 10 ng/ml TNFα for 24 h while the CTR group was stimulated with solvent only. (**G**) & (**H**) IL-8 and IL-1β expression in the intestinal organoid from the patient of inflammation bowel disease. The non-inflamed group represents the organoid derived from non-inflamed tissue and the inflamed group represents the organoid derived from inflamed tissue of the same patient.

pronounced in intestinal organoids not stimulated by TNFα (**Fig 1E and 1F**). In the context of IBD, the organoid derived from the inflamed tissue show similar cytokines expression pattern with Rifampicin treatment (**Fig 1G and 1H**). Thus PXR-mediated inhibition of pro-inflammatory gene expression in the inflamed intestine prominently involves the IEC compartment.

## Mutual repression of PXR and NF-κB signaling in IBD

Having established that the PXR can negatively regulate NF-kB-dependent gene transcription in IBD, we subsequently we decided to establish the potential relevance this observation. To this end, we determined *PXR* expression in our patient cohort and related this expression to NF-κB pathway activity as judged by *IL-8* and *IL-1ß* mRNA levels. We observed that expression of *PXR* is largely similar in control biopsies and in biopsies from quiescent CD patients. However, although not significant ($p$ = 0.15) *PXR* expression levels seemed to be reduced in biopsies from active CD patients (**Fig 2A**). When *PXR* expression was related to the expression of NF-κB target genes, we observed a statistically significant correlation between *PXR* expression and NF-κB activity (**Fig 2B**; $r$ = -0.6, $p$<0.01), suggesting that PXR status is important for controlling inflammation in IBD.

To further investigate this relationship between *PXR* expression and NF-κB signaling, patients were stratified into three groups according to their *PXR* mRNA expression level after Rifampicin treatment (**Fig 2C** and **S3 Fig**): patients that had lower *PXR* expression following Rifampicin ($n$ = 5), patients that had higher *PXR* expression following Rifampicin stimulation ($n$ = 10) and four patients that did not show changes in *PXR* expression following Rifampicin application. Using this stratification, it emerges that Rifampicin-mediated down regulation of *IL-8* expression correlates well with induction of PXR expression (**Fig 2D**; $p$<0.05), further highlighting the PXR-dependent nature of the anti-inflammatory action of Rifampicin in colonic biopsies. The overall upregulation of PXR expression seen in the Rifampicin-stimulated biopsies confirms the efficacy of stimulating PXR expression through the Rifampicin challenge (**Fig 2E**). It thus appears that the level of PXR activity is the rate-limiting factor with respect to NF-κB-directed gene expression in active IBD and conversely the amount of NF-κB activity is an important negative regulator for PXR expression. The latter notion was supported by observation made in intestinal organoids. PXR expression was repressed when NF-κB pathway TNFα stimulate in the intestinal organoids, although PXR still increased with Rifampicin treatment (**Fig 2F**). Rifampicin could also stimulate PXR expression in organoids deprived from IBD patient, but in the inflamed group the increase was held (**Fig 2G**). Thus it appears that PXR activation and NF-κB pathway are mutually exclusive in the context of the colon IEC.

## PXR mediates NF-κB inhibition

Direct support for the notion that PXR is important for restricting NF-κB activation in the epithelial compartment came from experiments in which we investigated the effect of *PXR*

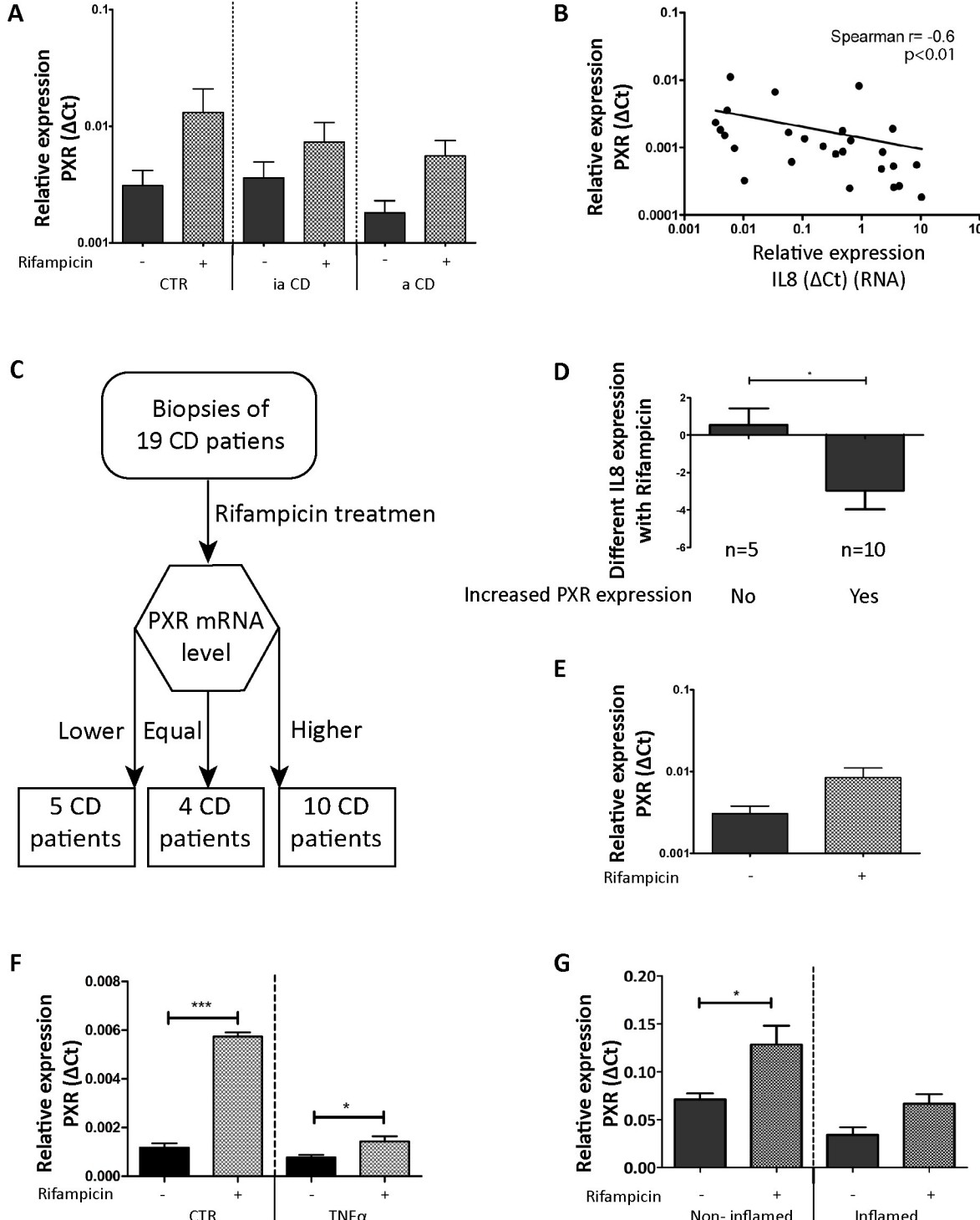

**Fig 2. PXR expression levels and relation to NF-κB activity.** (**A**) PXR mRNA levels in human intestinal biopsies. CTR are the biopsies from control patients ($n = 36$), ia CD signifies biopsies from CD patients without active intestinal inflammation ($n = 66$), and a CD indicates biopsies from CD patients with active intestinal inflammation ($n = 40$). The error bar is SEM. (**B**) Correlation between IL-8 mRNA levels and PXR mRNA levels in non-rifampicin-challenged biopsies. (**C**) Flowchart used for dividing CD patients into groups based on induction of PXR expression by Rifampicin treatment. The CD patients were divided into three groups: Group 1 (5 CD patients) show lower PXR expression after the Rifampicin treatment; Group 2 (4 CD patients) have equal expression of PXR before and after Rifampicin treatment; Group 3 (10 CD patients) have increased PXR expression after the Rifampicin treatment. (**D**) Effects of rifampicin on IL-8 expression as stratified by the effects of Rifampicin on PXR expression. The error bar is SEM, $^*p < 0.05$. (**E**) PXR expression in all biopsies

with or without Rifampicin stimulation. The error bar is SEM, $p$ = 0.056. (**F**) PXR mRNA expression in human intestinal organoids. The graph represents the mean PXR mRNA expression measure when stimulated with solvent only (0.1%(v/v) DMSO) or 100 μM Rifampicin for 18 h at 37 °C. The TNFα group was treated with 10 ng/ml TNFα for 24 h while the CTR group was stimulated with solvent only. The error bar is SEM. *$p$<0.05. ***$p$<0.001. (G) PXR mRNA expression in non-inflamed and inflamed intestinal organoids from IBD patient. The same methodology as in Fig 1G was used. The error bar is SEM. *$p$<0.05.

expression *per se* on NF-κB inhibition. We used the LS174t cells, a generally used model for colonic epithelial cells that recapitulates many aspects of normal enterocyte physiology [36] and generated two derivatives, the LS174t (siPXR) clone that lacked PXR expression and LS174t (nt) as a transfection control (**Fig 3A**). Consistently, induction of *CYP3A4* (the PXR target gene) was corrupted with PXR gene down-regulation (**Fig 3B**), but not in the control cells. Following stimulation of NF-κB with *E. Coli* lysate, IL-8 expression was enhanced in the cells lacking PXR as compared to controls, which was significantly higher than its expression in cells with PXR expression ($p$<0.05; **Fig 3C**). We confirmed this difference by showing a decreased p-p65 and p-Akt protein expression in the LS174t (nt) cell line (**Fig 3D** and **3E**). It demonstrated in this model cell line, PXR activity also constitutes the rate-limiting step in NF-κB-dependent gene expression. Conversely, activating NF-κB signaling reduces PXR levels in LS174t (nt) cells (**Fig 3F**). Thus these in vitro experiments showed that the negative relationship between NF-κB and PXR signaling is cell-autonomous and provide strong support for the notion that the presence of PXR represents an important target constraining NF-κB signalling in the mucosal epithelial compartment.

## Mutual repression of PXR and NF-κB signaling

In order to further understand the relationship between PXR and NF-κB signaling, we also measured the NF-κB target genes IL-8 and IL-1ß the in human epithelial colorectal adenocarcinoma cell line CACO2 [37]. As expected, NF-κB signaling was inhibited by Rifampicin treatment in TNFα-challenged monolayers (**Fig 4A** and **4B**). Conversely, induction of PXR by Rifampicin treatment was constrained in the presence of TNFα (**Fig 4C**). To establish that the rifampicin effects observed truly related to differences in NF-κB transcriptional activity we constructed a CACO2 clone containing a NF-κB reporter as described before [38]. The results show that also in this experimental system stimulation with rifampicin counteracts NF-κB activity (**Fig 4D**). In conclusion, PXR activity is the major rate-limiting pathway constraining mucosal NF-κB activity in active IBD and conversely active NF-κB signaling represses PXR expression. Thus targeting PXR emerges as a rational strategy for the management of IBD.

## Discussion

Active IBD is associated with the imbalanced immune response against intestinal microbial challenge. Xenobiotic and inflammatory signaling in response to microbiological constituents appear a certain extent mutually exclusive. Hence it is important to understand how xenobiotic receptor systems interact with epithelial immunity, especially in the context of IBD. Here we demonstrate that NF-κB signaling on one hand and the PXR receptor on the other hand restrain each other's activity and that in the context of active IBD the PXR pathway is a major rate-limiting factor for NF-κB-dependent epithelial gene expression (**Fig 4E**). Our observations have substantial consequences in our thinking of IBD and open the possibility that by targeting PXR signaling therapeutic benefit may be achieved, especially in those patients with epithelial hyper-activation of NF-κB signaling.

Earlier studies already demonstrated a role for defective xenobiotic resistance mechanisms in effector T cells for preventing Crohn's-like ileitis in experimental animals [39], the present

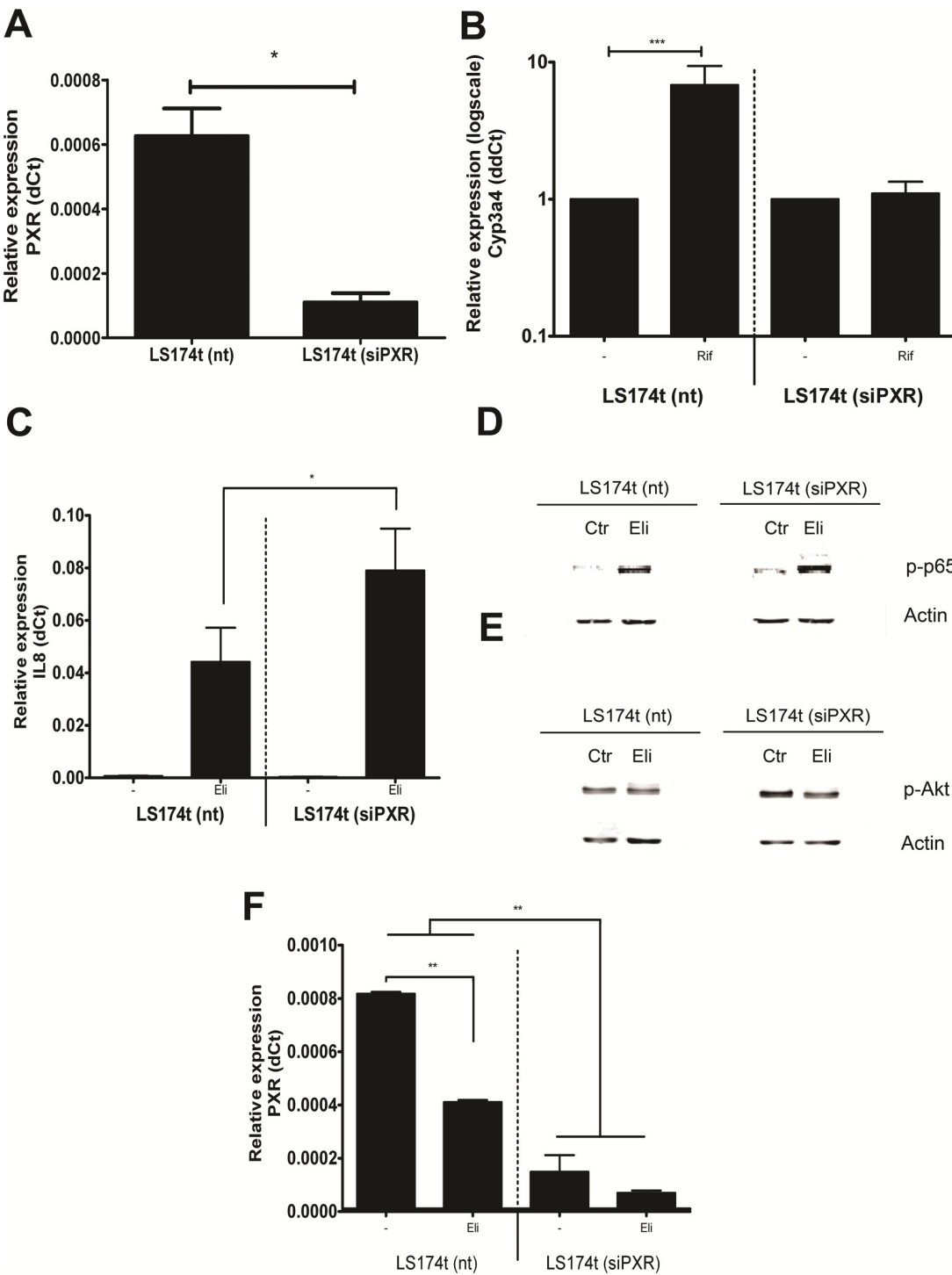

**Fig 3. Effects of PXR knock down on NF-κB signaling.** (**A**) PXR mRNA expression in a LS174t cell line stably transduced with a non-targeting siRNA (nt) or an siPXR. The graph represents PXR mRNA expression in LS174t cells from three independent experiments. The error bar is SD, *$p<0.05$. (**B**) LS174t (nt) and LS174t (siPXR) cells stimulated with 100 μM Rifampicin for 16 h at 37 °C. The relative mRNA expression of the PXR target gene *CYP3A4* is presented in the graph. The error bar is SD, ***$p<0.001$. (**C**) IL-8 mRNA expression in LS174t cells. Both cell lines were stimulated with 2 μl *E. coli* lysate (ELI). The error bar is SD, * $p<0.05$. (**D**) Activated NF-κB subunit p65 protein (p-p65) expression in LS174t cells. The same stimulation methods were used as in C. ß-actin protein expression is used as a loading control. (**E**) Activated Akt (p-Akt) protein expression in LS174t cells. (**F**) PXR mRNA expression in LS174t cells. The error bar is SD, ** $p<0.01$.

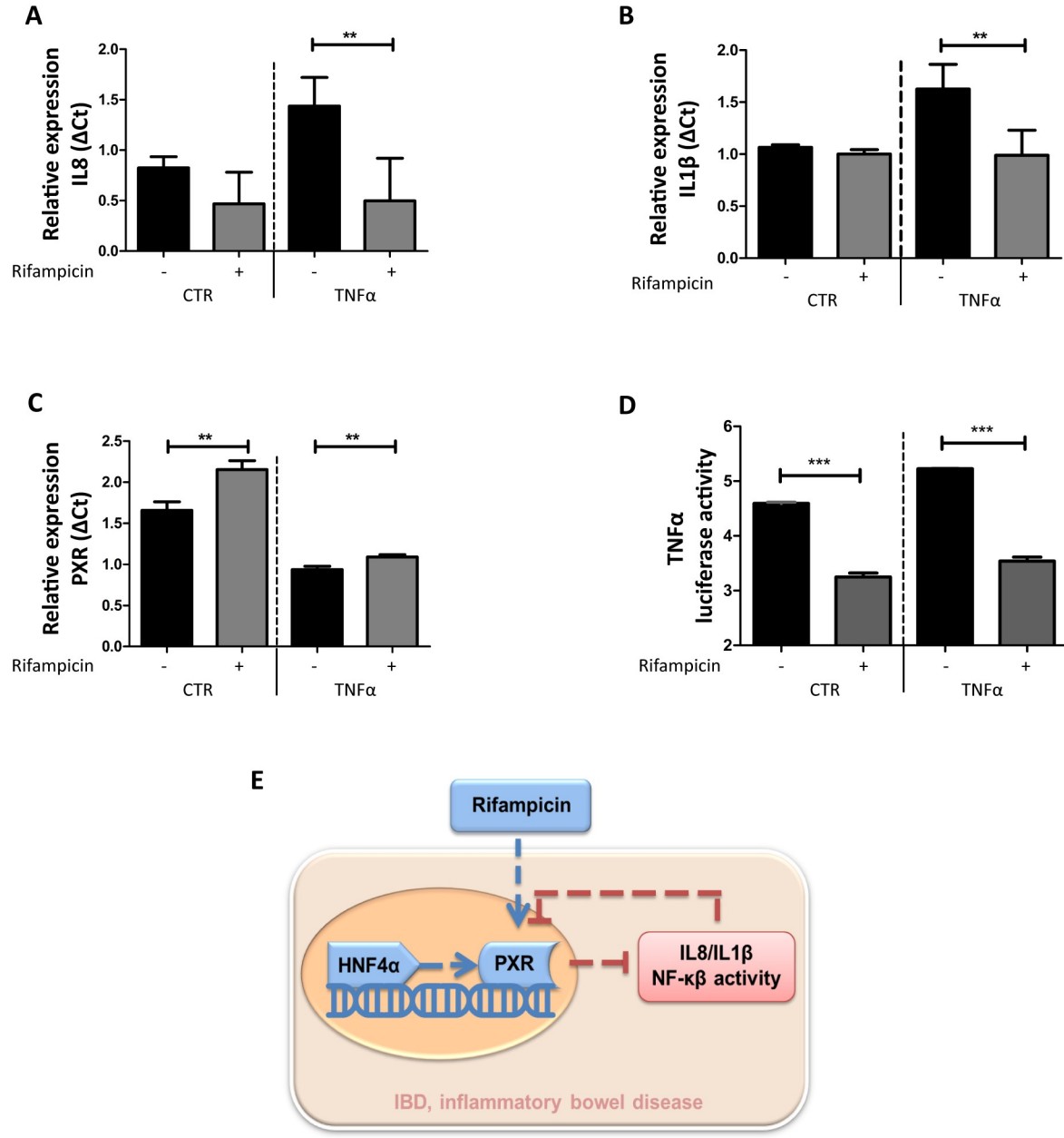

**Fig 4. PXR and NF-κB axis activity in CACO2 cells.** (**A**) &(**B**) IL-8 and IL-1β mRNA expression in CACO2 cells, respectively. CACO2 cells were stimulated with solvent only (0.1% (v/v) DMSO) or 100 μM Rifampicin for 16 h at 37 °C. The TNFα group was treated with 10ng/ml TNFα for 24 h while the CTR group was stimulated with solvent only. The error bar is SEM, and $^{**}p<0.01$. (**C**) PXR expression in CACO2 cells. (**D**) TNFα luciferase activity in CACO2 cells. The same stimulation methods were used as in A. (**E**) Schematic diagram illustrating the mutual repression of PXR and NF-κB.

study extends this concept into the epithelial compartment as well and indicates the importance of such mechanisms for active inflammatory responses. It is tempting to speculate why such mechanisms might exist, but a possibility is limiting NF-κB-dependent signaling and subsequently reduced inflammation facilitates regenerative responses. In apparent agreement with this notion is that Pregnane X receptor agonists enhance intestinal epithelial wound healing and repair of the intestinal barrier following the induction of experimental colitis [13].

Consistent with a critical role for PXR in constraining epithelial inflammatory responses are also the genetic studies that link genomic variation in PXR in susceptibility to IBD [11, 12], although a recent meta-analysis revealed that PXR gene polymorphisms may not be significantly associated with IBD susceptibility. It is, however, tempting to suggest that in certain patient populations aberrant PXR induction failed to control epithelial NF-κB induction and thus predisposing to disease. It would thus also be interesting to study the relation between such polymorphisms and the success of therapy to keep patients in remission, also in view of the association we see in the present study between PXR signaling and active disease. Larger studies containing cohorts are thus essential to clarify the association between *PXR* polymorphisms and the natural history of IBD. Disregarding the exact importance of genetic variance in the *PXR* gene, it is evident from the present study that PXR signaling constitutes a powerful anti-inflammatory mechanism capable of counteracting epithelial inflammation in active IBD. Stimulation of PXR may have clinical possibility, not only because of its capacity to limit inflammation, but also because such an action may improve bone mineralization [40, 41] and bone mineralization is a problem in inflammatory bowel disease [42, 43], whereas the lipophilic ligands used for PXR stimulation may conceivably also have chemopreventive effects with respect to the development of IBD-associated colorectal cancer [44]. We thus feel that our observation call for controlled studies assessing the potential of PXR agonists as a rational therapeutic strategy in IBD.

## Supporting information

**S1 Fig. The effect of rifampicin on PBMC.**
(DOCX)

**S2 Fig. Cancer cells with HNF4α stimulation.**
(DOCX)

**S3 Fig. The increase of PXR mRNA expression by rifampicin treatment.**
(DOCX)

**S4 Fig. The expression levels of PXR target genes in organoids.**
(DOCX)

**S5 Fig. The expression levels of PXR target genes.**
(DOCX)

**S1 Table. Primers used for qRT-PCR.**
(DOCX)

**S2 Table. Patients information with rifampicin.**
(DOCX)

**S3 Table. Patients information with linoleic acid.**
(DOCX)

**S4 Table. Patients information on biopsies.**
(XLS)

## Acknowledgments

We thank our patients for their consent and our coworkers for their support during this study.

## Author Contributions

**Conceptualization:** J. Jasper Deuring, Meng Li, Wanlu Cao, C. Janneke van der Woude, Maikel Peppelenbosch.

**Data curation:** J. Jasper Deuring, Meng Li, Wanlu Cao, C. Janneke van der Woude, Maikel Peppelenbosch.

**Formal analysis:** J. Jasper Deuring, Meng Li, Colin de Haar, Maikel Peppelenbosch.

**Funding acquisition:** Maikel Peppelenbosch.

**Investigation:** J. Jasper Deuring, Meng Li, Wanlu Cao, Sunrui Chen, Wenshi Wang, Colin de Haar, C. Janneke van der Woude.

**Methodology:** J. Jasper Deuring, Meng Li, Wanlu Cao, Sunrui Chen, Wenshi Wang, Colin de Haar, Maikel Peppelenbosch.

**Project administration:** C. Janneke van der Woude, Maikel Peppelenbosch.

**Resources:** C. Janneke van der Woude, Maikel Peppelenbosch.

**Supervision:** Maikel Peppelenbosch.

**Validation:** J. Jasper Deuring, Meng Li, Maikel Peppelenbosch.

**Visualization:** J. Jasper Deuring, Meng Li, C. Janneke van der Woude, Maikel Peppelenbosch.

**Writing – original draft:** J. Jasper Deuring, Meng Li, Colin de Haar.

**Writing – review & editing:** C. Janneke van der Woude, Maikel Peppelenbosch.

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
