## [Decision Letter · Decision Letter 0]

1 Jul 2019

PONE-D-19-14699

Pregnane X Receptor Activation Constrains Mucosal NF-�B Activity in Active Inflammatory Bowel Disease

PLOS ONE

Dear Dr. Peppelenbosch,

Thank you for submitting your manuscript to PLOS ONE. After careful consideration, we feel that it has merit but does not fully meet PLOS ONE’s publication criteria as it currently stands. Therefore, we invite you to submit a revised version of the manuscript that addresses the points raised during the review process.

We would appreciate receiving your revised manuscript by 8/30/2019. To enhance the reproducibility of your results, we recommend that if applicable you deposit your laboratory protocols in protocols.io, where a protocol can be assigned its own identifier (DOI) such that it can be cited independently in the future. For instructions see: http://journals.plos.org/plosone/s/submission-guidelines#loc-laboratory-protocols

We look forward to receiving your revised manuscript.

Kind regards,

Wenhui Hu, M.D., Ph.D.

Academic Editor

PLOS ONE

Journal Requirements:

Additional Editor Comments:

This is an interesting and significant study. It would be more interesting to show the cell types in the biopsies samples that express NFkB signaling or NFkB-dependent genes.

Reviewers' comments:

Reviewer's Responses to Questions

**Comments to the Author**

1. Is the manuscript technically sound, and do the data support the conclusions?

Reviewer #1: Yes

Reviewer #2: Yes

2. Has the statistical analysis been performed appropriately and rigorously? 

Reviewer #1: Yes

Reviewer #2: No

3. Have the authors made all data underlying the findings in their manuscript fully available?

Reviewer #1: No

Reviewer #2: Yes

4. Is the manuscript presented in an intelligible fashion and written in standard English?

Reviewer #1: Yes

Reviewer #2: Yes

5. Review Comments to the Author

Reviewer #1: The study investigates the importance of PXR signaling in inhibiting NF-kB activity in the intestinal epithelial compartment during the IBD. Using colonic biopsies from control or IBD patients and several cell culture models, the authors observed a strict inverse correlation between colonic epithelial PXR levels and NF-kB target gene expression. The results are consistent with the well-established crosstalk between PXR and NF-kB signaling pathways. However, this study lacks mechanistic insight and did not provide further information on the known crosstalk. In addition, figure legends were misplaced, and grammar errors and typos were found throughout the manuscript.

Specific comments:

1. As mentioned in the method “Biopsies”, the biopsies were taken from the ascending colon, the transversum and the descending colon. Do these parts of colon show different expression patterns of PXR or activation pattern of NF-kB? Which part did the authors choose for the further study like “Stimulation of the biopsies”?

2. To further investigate the crosstalk between PXR and NF-kB signaling, patients were stratified into three groups according to their PXR mRNA expression following Rifampicin treatment. The detailed information about PXR expression levels should be provided.

3. When the authors investigate the effects of NF-kB signaling on the PXR expression and activation, the expression levels of more PXR target genes should be measured.

4. IL-1β protein levels were not shown in Figure 1 but was mentioned in line 190.

5. Missing labels in Figure 2 panel F.

6. Some typos were also found: e.g. “Rifamcipin” in line 131; “37 。C” in line 278.

Reviewer #2: This is an interesting study, demonstrating that PXR levels are inversely correlated with NF-kappa B activity in the intestinal epithelial cells, conversely NF-kappa B signaling down-regulates PXP expression. Human tissues, intestinal organoids, and cell culture models were studied. The study suggests that PXR may have the potential to be a therapeutic target to inhibit NF-kappa B activity and gut inflammation. I have a few comments.

1. The abstract is not very informative. It would be helpful to expand the Results section of the Abstract. If word limit is a concern, shorten the Methods section.

2. Fig. 2B: is p value > 0.01 or < 0.01? Fig. 2 F, the x-axis label is missing?

3. The description of Fig. 3 (Lines 285 to L295) appears not clear. Please reconcile the description with the figure.

4. Please check mistakes through the paper, i.e. L69-L70; L202; L241.

6. PLOS authors have the option to publish the peer review history of their article (what does this mean?). If published, this will include your full peer review and any attached files.

Reviewer #1: No

Reviewer #2: No

---

## [Author Response · Author response to Decision Letter 0]

25 Jul 2019

Journal Requirements:

We had revised our manuscript and supplementary files according to the journal requirements.

The captions for supporting information files were included at the end of our manuscript.

Due to the PXR staining of the endoderm are not core part of our study, we removed the sentence refer to the data.

Additional Editor Comments:

This is an interesting and significant study. It would be more interesting to show the cell types in the biopsies samples that express NFkB signaling or NFkB-dependent genes.

The human protein atlas (https://www.proteinatlas.org/) show the protein expression of NF-κβ in colon tissue.

Cell types NF-κβ protein expression 

Endothelial cells Low

Glandular cells Medium

In the tissue of IBD patients, previous studies indicate NF-κβ is mostly found in mucosal macrophages and epithelial cells 1, 2, 3.

Immunofluorescence detection of activated NF-κB in the inflamed mucosa. Activated NF-κB (Cy3 fluorescence) was identified in the lamina propria or in crypts near the basal membrane in inflamed mucosa. In normal mucosa, only nonspecific autofluorescence was detected. (A) Normal mucosa. The crypts show autofluorescence (orange). No red fluorescence (activated NF-κB, Cy3) can be detected. (B) Diverticulitis. Specific red fluorescence is detectable mainly in the lamina propria (arrowheads), corresponding to nuclei of cells containing activated NF-κB. (C) Crohn's disease mucosa without macroscopic inflammation (score 0). Only autofluorescence can be seen. The autofluorescence in the lamina propria seems to be greater compared with (A) control specimens perhaps because of the deposition of collagen. (D) Crohn's disease with a low degree of inflammation (+). Activated NF-κB can be detected (arrowheads). (E) Ulcerative colitis without inflammation (0). Again mainly autofluorescence is visible. (F–H) Ulcerative colitis in a severe (+++) state of inflammation. A large number of stained nuclei is detectable (arrowheads). Positive cells are located in the lamina propria and in the crypts (original magnification: A–F, 400×; G, 250×; and H, 1000×). NF-κB activation in uninflamed and inflamed areas of the same patients. Specimens were prestained only with eosin. Activated NF-κB was stained with the α-p65mAb and BDHC (blue granular reaction product). In uninflamed mucosa, the number of activated cells was clearly lower. An obviously greater number of activated cells (arrows) is present in the inflamed areas. (A) Nonspecific colonic inflammation (inflammation grading, 0). (B) Nonspecific colonic inflammation (inflammation grading, +). Activated NF-κB is clearly present in the basal nuclei of crypt epithelial cells (basal membrane marked). (C) Crohn's disease (inflammation grading, 0). (D) Crohn's disease (inflammation grading, +++). Activated NF-κB is found close to small blood vessels (red-stained erythrocytes). (E) Ulcerative colitis (inflammation grading, 0). (F) Ulcerative colitis (inflammation grading, +++). Activated NF-κB is again visible close to small blood vessels.

 Joo Sung Kim et al. in their study mentioned the different characteristics of high and low NF-κβ activity patient groups (table below).

Han YM et al. (2017) NF-kappa B activation correlates with disease phenotype in Crohn's disease. PLoS ONE 12(7): e0182071.

 When staining PXR in biopsies (not the same biopsies that we used for the stimulation experiments), the inflamed intestinal tissue show relatively low PXR expression (see below). Interestingly, the PXR expression is mainly found in epithelial cells.

Ctr is a biopsy from healthy individual; IA is inactive IBD; A is active IBD; ISC is ischemic colitis, IFC is infectious colitis; SB is small bowel

Review Comments to the Author

Reviewer #1: The study investigates the importance of PXR signaling in inhibiting NF-kB activity in the intestinal epithelial compartment during the IBD. Using colonic biopsies from control or IBD patients and several cell culture models, the authors observed a strict inverse correlation between colonic epithelial PXR levels and NF-kB target gene expression. The results are consistent with the well-established crosstalk between PXR and NF-kB signaling pathways. However, this study lacks mechanistic insight and did not provide further information on the known crosstalk. In addition, figure legends were misplaced, and grammar errors and typos were found throughout the manuscript.

Specific comments:

1. As mentioned in the method “Biopsies”, the biopsies were taken from the ascending colon, the transversum and the descending colon. Do these parts of colon show different expression patterns of PXR or activation pattern of NF-kB? Which part did the authors choose for the further study like “Stimulation of the biopsies”?

The different parts of colon show similar expression patterns of PXR and NF-κβ. Mostly the ascending colon and the transversum colon were used for further study and if they are not available, terminal ileum for small bowel would be taken instead. The detail information was shown in supplementary table 4.

2. To further investigate the crosstalk between PXR and NF-kB signaling, patients were stratified into three groups according to their PXR mRNA expression following Rifampicin treatment. The detailed information about PXR expression levels should be provided.

We show the detailed information of PXR expression level with Rifampicin treatment in supplementary file (Fig S3).

3. When the authors investigate the effects of NF-kB signaling on the PXR expression and activation, the expression levels of more PXR target genes should be measured.

Cyp3a4 and more genes were measured in supplementary figure 4 and supplementary figure 5.

And we also investigated the crosstalk between PXR and NF-kB signalling directly in the organoid derived from IBD patients (Fig 1G and 1H) (Fig 2G).

4. IL-1β protein levels were not shown in Figure 1 but was mentioned in line 190.

We measured protein level of IL-8 and mRNA level of IL-1β. Therefore we removed the phrase related with IL-1β protein levels. 

5. Missing labels in Figure 2 panel F.

Labels of Figure 2F had been added.

6. Some typos were also found: e.g. “Rifamcipin” in line 131; “37 。C” in line 278.

Mistakes were corrected.

Reviewer #2: This is an interesting study, demonstrating that PXR levels are inversely correlated with NF-kappa B activity in the intestinal epithelial cells, conversely NF-kappa B signaling down-regulates PXP expression. Human tissues, intestinal organoids, and cell culture models were studied. The study suggests that PXR may have the potential to be a therapeutic target to inhibit NF-kappa B activity and gut inflammation. I have a few comments.

1. The abstract is not very informative. It would be helpful to expand the Results section of the Abstract. If word limit is a concern, shorten the Methods section.

Abstract was revised to expand the results section to reveal more information.

2. Fig. 2B: is p value > 0.01 or < 0.01? Fig. 2 F, the x-axis label is missing?

Figure 2B is p<0.01 and were corrected. Labels of Figure 2F had been added.

3. The description of Fig. 3 (Lines 285 to L295) appears not clear. Please reconcile the description with the figure.

We reorganized the description of Fig 3.

4. Please check mistakes through the paper, i.e. L69-L70; L202; L241.

Mistakes were corrected.

 

Reference

1. Atreya I, Atreya R, Neurath MF. NF-kappaB in inflammatory bowel disease. J Intern Med 2008, 263(6): 591-596.

2. Rogler G, Brand K, Vogl D, Page S, Hofmeister R, Andus T, et al. Nuclear factor kappaB is activated in macrophages and epithelial cells of inflamed intestinal mucosa. Gastroenterology 1998, 115(2): 357-369.

3. Han YM, Koh J, Kim JW, Lee C, Koh SJ, Kim B, et al. NF-kappa B activation correlates with disease phenotype in Crohn's disease. PLoS One 2017, 12(7): e0182071.

---

## [Decision Letter · Decision Letter 1]

20 Aug 2019

Pregnane X Receptor Activation Constrains Mucosal NF-�B Activity in Active Inflammatory Bowel Disease

PONE-D-19-14699R1

Dear Dr. Peppelenbosch,

We are pleased to inform you that your manuscript has been judged scientifically suitable for publication and will be formally accepted for publication once it complies with all outstanding technical requirements.

With kind regards,

Wenhui Hu, M.D., Ph.D.

Academic Editor

PLOS ONE

Additional Editor Comments (optional):

Reviewers' comments:

Reviewer's Responses to Questions

**Comments to the Author**

1. If the authors have adequately addressed your comments raised in a previous round of review and you feel that this manuscript is now acceptable for publication, you may indicate that here to bypass the “Comments to the Author” section, enter your conflict of interest statement in the “Confidential to Editor” section, and submit your "Accept" recommendation.

Reviewer #1: All comments have been addressed

Reviewer #2: All comments have been addressed

2. Is the manuscript technically sound, and do the data support the conclusions?

Reviewer #1: Yes

Reviewer #2: Yes

3. Has the statistical analysis been performed appropriately and rigorously? 

Reviewer #1: Yes

Reviewer #2: Yes

4. Have the authors made all data underlying the findings in their manuscript fully available?

Reviewer #1: No

Reviewer #2: Yes

5. Is the manuscript presented in an intelligible fashion and written in standard English?

Reviewer #1: Yes

Reviewer #2: No

6. Review Comments to the Author

Reviewer #1: the authors have addressed most major concerns. The revised manuscript is now acceptable for publication at PLOS One.

Reviewer #2: The revision has addressed all my questions raised in the original review. I have no further comments.

7. PLOS authors have the option to publish the peer review history of their article (what does this mean?). If published, this will include your full peer review and any attached files.

Reviewer #1: No

Reviewer #2: No

---

## [Editor Report · Acceptance letter]

25 Sep 2019

PONE-D-19-14699R1 

Pregnane X Receptor Activation Constrains Mucosal NF-κB Activity in Active Inflammatory Bowel Disease 

Dear Dr. Peppelenbosch:

I am pleased to inform you that your manuscript has been deemed suitable for publication in PLOS ONE. Congratulations! Your manuscript is now with our production department. 

With kind regards,

on behalf of

Dr. Wenhui Hu 

Academic Editor

PLOS ONE